# Swine Enteric Coronavirus: Diverse Pathogen–Host Interactions

**DOI:** 10.3390/ijms23073953

**Published:** 2022-04-02

**Authors:** Quanhui Yan, Xiaodi Liu, Yawei Sun, Weijun Zeng, Yuwan Li, Feifan Zhao, Keke Wu, Shuangqi Fan, Mingqiu Zhao, Jinding Chen, Lin Yi

**Affiliations:** 1College of Veterinary Medicine, South China Agricultural University, Guangzhou 510642, China; yanqh@stu.scau.edu.cn (Q.Y.); lxd18839462378@163.com (X.L.); syw18530494979@stu.scau.edu.cn (Y.S.); zwj1102727352@stu.scau.edu.cn (W.Z.); waner20191028012@stu.scau.edu.cn (Y.L.); zhaofeifan1234@163.com (F.Z.); 13660662837@163.com (K.W.); shqfan@scau.edu.cn (S.F.); zmingqiu@scau.edu.cn (M.Z.); 2Guangdong Laboratory for Lingnan Modern Agriculture, College of Veterinary Medicine, South China Agricultural University, Guangzhou 510642, China; 3Key Laboratory of Zoonosis Prevention and Control of Guangdong Province, Guangzhou 510642, China

**Keywords:** coronavirus, host–virus interaction, autophagy, apoptosis, innate immunity

## Abstract

Swine enteric coronavirus (SeCoV) causes acute gastroenteritis and high mortality in newborn piglets. Since the last century, porcine transmissible gastroenteritis virus (TGEV) and porcine epidemic diarrhea virus (PEDV) have swept farms all over the world and caused substantial economic losses. In recent years, porcine delta coronavirus (PDCoV) and swine acute diarrhea syndrome coronavirus (SADS-CoV) have been emerging SeCoVs. Some of them even spread across species, which made the epidemic situation of SeCoV more complex and changeable. Recent studies have begun to reveal the complex SeCoV–host interaction mechanism in detail. This review summarizes the current advances in autophagy, apoptosis, and innate immunity induced by SeCoV infection. These complex interactions may be directly involved in viral replication or the alteration of some signal pathways.

## 1. Introduction

Coronaviruses (CoVs) are enveloped viruses with a positive-sense, single-stranded RNA genome. Some coronaviruses can infect hosts and cause respiratory damage, such as SARS-CoV, MERS-CoV, and SARS-CoV-2 [1]. As the COVID-19 epidemic is sweeping the world, many studies have also focused on the interaction mechanism between SARS-CoV-2 and the host [2]. Notably, certain coronaviruses are now known to trigger antibody-dependent enhancement (ADE) to enhance infection, such as SARS-CoV-2 and feline coronavirus (FCoV) [3]. However, to date, there are no studies on whether swine enteric coronavirus (SeCoV) can trigger ADE, as there are relatively few studies on SeCoV.

Coronaviruses are members of the *Nidovirales* order, family *Coronaviridae*, subfamily *Coronavirinae* (International Committee on Taxonomy of Viruses). *Coronavirinae* can be divided into four genera, *Alpha-*, *Beta-*, *Gamma-* and *Delta**coronaviruses*, according to serological and genomic methods. Known members of the SeCoV family include porcine epidemic diarrhea virus (PEDV), transmissible gastroenteritis virus (TGEV), porcine delta coronavirus (PDCoV), and swine acute diarrhea syndrome coronavirus (SADS-CoV) [4]. Among the four known SeCoVs, PDCoV belongs to *Deltacoronavirus*, whereas PEDV, TGEV, and SADS-CoV belong to *Alphacoronavirus* (Figure 1).

Among them, TGEV has been circulating in pigs for decades since it broke out in the 1940s. As early as 1933, the state of Illinois in the United States had related records of TGEV [5]. TGEV was first reported in the United States in 1946 and later in Europe, Asia, Africa, and South America [6]. The infection of pigs caused by TGEV is mainly through the mouth, nose, and mucosa of pigs. When there were virions in the external environment, the viruses entered the respiratory system from the nasal cavity and multiplied in large numbers [7]. Then, TGEV invaded the intestinal epithelium in the small intestine of pigs with blood circulation and replicated; severe villus contraction and malabsorption led to diarrhea. The mortality rate of defenseless piglets after infection was almost 100% [8]. Clinically diseased pigs and infected pigs excrete virions from feces, vomit, and exhale gas to transmit TGEV through the respiratory and digestive tract.

The length of the genomic sequence of TGEV is about 28.5 kb [9]. The S protein encoded by its *S* gene, similar to other coronaviruses, determines the tissue tropism and virulence of TGEV. It is worth mentioning that TGEV has evolved a new type of *Alphacoronavirus* called PRCV. Compared to TGEV, 621–681 bp is shortcoming from the N-terminus of the *S* gene of PRCV. At the same time, its histophilicity has been changed from intestinal to respiratory, and its virulence has been reduced [10]. 

PEDV was first isolated and identified in Europe about 40 years ago and has been prevalent in Eurasia since the 1990s [11,12]. It broke out in China around 2010 and became a significant swine viral disease [13,14,15]. The clinical symptoms and pathological changes caused by PEDV infection are almost the same as those of TGEV, which leads to host vomiting, malabsorption, diarrhea, and other symptoms [16,17]. However, the difference is that TGEV causes disease in pigs of all ages, while unweaned piglets aged 4–5 days during the PEDV subtype I epidemic in 1972 (Oldham 1972) were less affected. After the continuous evolution and variation of PEDV subtype I, PEDV subtype II occurred in 1976, leading to varying degrees of diarrhea in pigs of all ages. The severity and mortality of infected pigs may be inversely proportional to the age of pigs. The incidence of piglets infected with PEDV is close to 100%, while the incidence of sows may be different. Compared to weaned piglets, PEDV causes more severe diseases in suckling piglets, but the reason is not apparent. Slow intestinal cell regeneration may be essential in suckling piglets.

Similar to other coronaviruses, the genome size of PEDV is about 28 kb. Compared to HCoV-229E (GenBank: KU291448), HcoV-NL63 (GenBank: LC687394), and TGEV, the CV777 strain (GenBank: AF353511) of PEDV subtype I and bat coronavirus BtCoV/512/2005 (GenBank: DQ648858) share common evolutionary precursors in phylogeny and the genome structure [18]. The RdRp gene of coronavirus UNICAMP_bat_BR_14 (GenBank: KM514667) from bats is closely related to PEDV [19]; there may be a cross-species transmission of coronavirus between bats and pigs. On the other hand, PEDV can replicate in the primary target cells of natural hosts and effectively infect human, monkey, and bat cells [20]. Therefore, it is speculated that PEDV may be transmitted by bat coronavirus, which initially causes diarrhea in fattening pigs and adult pigs and is then acquired by piglets.

*Deltacoronavirus* (DCoV) is a new coronavirus genus identified by The International Committee on Taxonomy of Viruses (ICTV) 2011. DCoV was first detected in Chinese ferret badgers and Asian leopard cats in 2007 [21,22]. Porcine coronavirus was a newly discovered porcine coronavirus in 2012. It has been confirmed that PDCoV has strong pathogenicity and causes watery diarrhea and vomiting in sows and piglets. The mortality rate of suckling piglets after infection reached 30% to 40% [23]. The clinical symptoms caused by PDCoV infection were similar to those of PEDV and TGEV, so PDCoV was not paid attention to when it first appeared, and its harm to newborn piglets was ignored [24]. In February 2014, the Department of Agriculture of Ohio examined the fecal or intestinal samples of 42 diarrhea sows and piglets from 5 farms in the state. The detection rate of PDCoV was as high as 92.9% [25]. Since then, PDCoV infection has broken out widely in the United States, and PDCoV has also appeared in Canada, South Korea, Thailand, and other places [26,27,28]. Among the four known genera of *Coronaviridae*, human infections reported to date are limited to *Alphacoronavirus* and *Betacoronavirus* [29,30,31]. However, recent research identified PDCoV strains in plasma samples of three Haitian children with acute undifferentiated fever in 2021, suggesting that evolutionary changes and adaptation may lead to cross-species transmission of coronavirus [32].

Similar to PEDV and TGEV, SADS-CoV belongs to *Alphacoronavirus*. It was an important pathogen that caused the infection and death of newborn piglets found in Guangdong Province, China, in January 2017. Similar to PDCoV, SADS-CoV is a new type of SeCoV that has only been discovered recently. SADS-CoV causes vomiting, watery diarrhea, dehydration, and death in newborn piglets under 5 days old, with a mortality rate of up to 90%. The mortality of piglets over 8 days old decreased significantly after infection, and the sows showed only mild diarrhea [33]. A retrospective survey of SADS-CoV in 45 pig farms in Guangdong Province was conducted based on 236 diarrhea samples. The results showed that SADS-CoV first appeared in Guangdong Province in August 2016 [34]. SADS-CoV is closely related to bat coronavirus. The sequence of SADS-CoV genome detected from intestinal samples of diarrhea piglets from pig farms in Guangdong Province is more than 90% similar to bat coronavirus HKU2-CoV (GenBank: NC_009988.1) [33]. HKU2-CoV was isolated from bats in 2016, speculating that SADS-CoV was transmitted from bats to piglets across species. To date, the virus has not been reported in other countries (Table 1).

## 2. SeCoV Infection and the Involvement of Host Factors

### 2.1. Morphology and Genomic Structure of SeCoV

Coronaviruses are spherical or oval, with a diameter of 80–120 nm [35]. Most coronaviruses have coronal processes formed by the trimeric spike (S) protein, and the surface protuberances of some special coronaviruses (such as HCoV-HKU1) are composed of dimer hemagglutinin-esterase (HE) proteins [36]. The process on the surface involves the S protein, which is a type I transmembrane protein, and the extracellular domain is larger than the intracellular domain. The S protein mainly mediates virus invasion and determines viral tissue or host tropism by binding to host cell receptors. The envelope is primarily composed of the membrane (M) protein and has three transmembrane domains [37]. The low content of the envelope (E) protein also exists in the envelope, and the E and M proteins are mainly involved in the assembly process of the virus [38]. The nucleocapsid protein wraps the virus genome to form a nucleoprotein complex with a helical symmetrical structure [37]. The total genome length of coronavirus reaches 27–32 kilobases, which is the largest of all the known RNA viruses. Its genome is a single-stranded, positive-sense RNA that is 5′-capped and 3′-polyadenylated, containing 6–10 open reading frames (ORFs). The genomic sequence of SeCoV is 5′UTR-ORF1a-ORF1b-S-E-M-NS6-N-NS7-3ʹUTR (Table 1). ORF1 (consisting of overlapping ORF1a and ORF1b) encodes viral replicase, accounting for 2/3 of the SeCoV genome. Nsp12 with RNA-dependent RNA polymerase (RdRP) activity mediates the replication and transcription of viral genome, resulting in genomic RNA (gRNA) and subgenomic RNA (sgRNA) [39,40]. The short sgRNA encodes conserved structural proteins, including S protein, E protein, M protein, and N protein, and several auxiliary proteins [41] (Figure 2).

The replication cycle is as follows: the virus first attaches to the cell surface and enters the cell, then the viral replicase translates, then the viral genome is transcribed and replicated, and the virions are assembled and released after the structural protein translation.

### 2.2. Attachment and Entry

The first step in the entry of SeCoVs into the host is to bind to the receptors of susceptible cells through the S protein. The extracellular domain of the S protein is divided into two different functional domains: S1 and S2 [42]. The S1 region is responsible for encoding the part in which the virus binds to the cell receptor; the S2 region contains fusion peptides that help the virus to fuse with the cell surface. In the S1 region, two independent folding regions are responsible for binding to the receptor cells: the N-terminal domain (NTD) and the C-terminal domain (CTD). NTD has the activity of binding to sialic acid, and CTD can bind to cellular protein receptors [43]. When the S1 region of S protein binds to the receptor, the conformation of S protein changes and the deep cleavage site is exposed to the surface. Subsequently, protease cleavage leads to the separation of S1 and S2, which helps the fusion peptide in S2 insert into the host cell membrane and leads to membrane fusion [44,45,46,47,48].

The binding of the S protein to receptor is the primary determinant of host range and tissue tropism of CoV. In this process, the S protein of CoV interacts with the corresponding host factors to realize the adsorption and entry of target cells. At present, four types of CoV functional protein receptors have been identified: aminopeptidase-N (APN), carcinoembryonic antigen-related cell adhesion molecule 1 (CEACAM1), angiotensin-converting enzyme 2 (ACE2), and dipeptidyl peptidase 4 (DDP4). Some *Betacoronavirus* use other kinds of functional receptors, such as CEACAM1 for mouse hepatitis virus (MHV), ACE2 for severe acute respiratory syndrome coronavirus (SARS-CoV) [49], and DDP4 for Middle East respiratory syndrome coronavirus (MERS-CoV) [50].

Some SeCoVs use enzymes on the cell surface as receptors. For example, TGEV from *Alphacoronavirus* uses pAPN as its functional receptor (Table 1). TfR1 is widely distributed in the surface epithelial cells of anemic newborns, and the intestinal epithelial cells of newborn piglets are the targets of TGEV. TfR1 enhances the invasion of TGEV to cells, and the S1 region of the TGEV S protein interacts with the extracellular domain of TfR1, so TfR1 is likely to be the second receptor of TGEV [51]. TGEV is not inactivated when passing through stomach acid, while PRCV, a mutant of TGEV, is not acid-resistant, partly because of whether it can bind to SA [52]. There is an SA binding site on the S protein of TGEV, which can bind to the mucin of the brush border membrane and help the virion pass through the mucous layer and the cell envelope of the top membrane of the intestinal cell, and finally reach the top membrane of the intestinal cell [53,54]. In the early stage of TGEV infection, EGFR, as a cofactor of TGEV invasion, cooperates with pAPN to activate PI3K/AKT and MEK/ERK1/2 signal pathways to promote the entry of TGEV into cells through endocytosis [55].

The cellular receptors of PEDV and SADS-CoV, both *Alphacoronavirus* members, are not clear. Previous studies have shown that PEDV could successfully infect through overexpressing pAPN in Madin-Darby canine kidney (MDCK) cells. It is preliminarily determined that the functional receptor of PEDV is pAPN [56], Neu5Ac acts as a coreceptor [20]. In addition, the transgenic mouse model expressing pAPN is susceptible to PEDV, which further proves that pAPN is the receptor of PEDV [57]. However, the results of later studies overturned this conclusion, so pAPN is not a functional receptor of PEDV. Nonetheless, through its protease activity to promote PEDV infection [58], the fact that PEDV is infected with APN-knockout pigs confirmed this [59,60,61]. PEDV infects the cell lines of pigs, monkeys, humans, bats, and other species, further indicating that the interaction between PEDV and APN is not specific and suggests that PEDV can spread across species [20]. In addition to APN, the NTD of PEDV S1 has an SA-binding activity. The combination of S1 NTD with SA on the cell surface may enhance the infectivity of the virus [62,63]; the mechanism of how the binding of SA and the S protein affects the entry of PEDV into cells is unclear. In addition, EGFR plays a positive role in the entry of PEDV into cells [64].

Since SADS-CoV has only been discovered in recent years, relatively few studies on its binding receptors exist [65,66]. It has been reported that the main target of SADS-CoV infection was the host splenic dendritic cells (DC) and did not use the known CoV receptor to enter the cell [67].

PDCoV is currently the only SeCoV member of *Deltacoronavirus*, and there is no conclusion on the results of receptor research. Recently, it has been reported that there is a disagreement as to whether pAPN is the receptor of PDCoV invading host cells. The exogenous expression of pAPN can successfully infect PDCoV in non-infectious cell lines [68], reflecting the origin of PDCoV and the mechanism of its cross-species transmission. However, pAPN knockout or treatment with APN-specific antibodies and inhibitors only reduces PDCoV infection to a certain extent, so pAPN is not a functional receptor of PDCoV [69]. Due to the fact that PDCoV agglutinates rabbit red blood cells, neuraminidase (NA) inhibits erythrocyte agglutination, suggesting that SA inhibited by NA may be a cofactor of PDCoV infection [70]. SA can promote the adhesion ability of PDCoV, relating to the intestinal tissue tropism of PDCoV. These results suggest that SA may be a cofactor of PDCoV infection [71].

According to the current research results, there are two ways for CoVs to invade cells; one is endocytosis. Viral S protein interacts with receptor and then virus particles are wrapped in endosomes to enter host cells [72]. In endosomes, the viral envelope and membrane structure of endosomes are fused in the environment of cathepsin L (CTSL) and low pH to release the viral genome into the host cytoplasm [73]. The other is membrane fusion, through which exogenous proteases or host proteases activate the S protein upon the binding of the virus to the receptor, and that virus then enters the cytoplasm through membrane fusion.

Host proteases play a crucial role in virus infection. Different viruses interact with different proteases to enter the cell, which determines the invasion pathway of the virus. Four proteases are involved in the process of coronavirus infection: membrane-binding proteases (e.g., transmembrane serine protease), lysosomal proteases (e.g., CTSL), extracellular proteases (e.g., trypsin), and proprotein convertases (e.g., Furin) [74]. For example, transmembrane serine protease 2 (TMPRSS2) and mosaic long serine protease (MSPL) cleave the S protein to promote the fusion of PEDV and cells [75]. Similar to PEDV, CTSL and cathepsin B (CTSB), these two lysosomal proteases activate the S protein of PDCoV to enter the cell through endocytosis [74,76]. As an extracellular protease, trypsin activates the PDCoV S protein to induce intercellular fusion. Trypsin is needed in the culture and passage of SeCoV, which promotes the fusion of virus and cells and increases the yield of virus [71,77,78,79] (Figure 3).

Some host factors also limit the attachment of SeCoVs. For example, interferon-induced transmembrane proteins (IFITMs) reveal a broad-spectrum antiviral effect [80,81,82]. IFITMs regulate cell membrane properties to prevent the virus from entering through membrane fusion [83,84]. IFITMs inhibit the invasion mediated by SARS-CoV S protein, and this process is affected by exogenous cathepsin [85]. Therefore, IFITM has a significant antiviral effect on viruses, such as SeCoVs, which invade host cells through endocytosis and endosomal acidification [81]. For example, IFITM1 and IFITM3 induced by interferon-λ (IFN-λ) can inhibit the infection of porcine intestinal epithelial cells (IPECs) by PEDV [86]. In contrast, IFITM2 or IFITM3 promote HCoV-OC43 infection in host cells [87]. IFITM2 interacts with the NTD of SARS-CoV-2 S protein to promote the fusion of SARS-CoV-2 in the early endosome, thus promoting virus infection [82]. Different IFITM family proteins show different inhibitory abilities against different viruses [88]. IFITM1 inhibits the replication of coronavirus more significantly than IFITM3 [85]. Recent studies identified some structural motifs of IFITM, which determine the antagonistic or promotive activity of IFITM against viral infections [89]. The interaction mechanism between host factors, such as IFITM and SeCoVs, will be a new direction for future research.

## 3. Autophagy Induced by SeCoV Infection

Macroautophagy (subsequently referred to as autophagy) is an essential process of evolutionarily conserved turnover of intracellular substances in eukaryotes, maintaining the self-stability of macromolecules, energy, and organelles [90,91]. Especially in cells under stress conditions (such as hunger, growth factor deficiency, and pathogen infection), autophagosomes (APs) encapsulate damaged proteins or organelles and fuse with lysosomes to form autolysosomes (ALs) for degradation and recycling [1]. The process of autophagy is regulated by conservative autophagy-related genes (ATGs). After autophagy, cytoplasmic microtubule-associated protein light chain 3 (LC3-I) is lipided to the form of LC3-II; the ratio of LC3-II/LC3-I is an indicator of autophagy activity [92] (Figure 4).

Previous studies have shown that autophagy gives the body innate protection against pathogens, because the degradation process of autophagy could kill pathogens and present them to the immune system [93,94,95]. Autophagy can also have a positive or negative effect on virus replication, depending on the nature of the virus. Autophagy plays an anti-viral role during the infection of herpesvirus HSV-1, whereas adenovirus, porcine circovirus type 2 (PCV2), and certain other DNA viruses can use autophagy to promote replication [96]. Similar to other RNA viruses, some coronaviruses can use autophagy to promote their replication [97,98,99,100]. It is reported that mouse hepatitis virus (MHV) can use autophagy to promote its replication [101], but other studies yielded the opposite results [102].

The relationship between SeCoVs and autophagy has not been fully characterized; the effect of autophagy on the replication of SeCoVs is also different in different cells (Table 1). TGEV infection can induce mitochondrial autophagy (mitophagy) in IPEC cells mediated by DJ-1 (a multifunctional redox-sensitive protein), which enhances this replication by reducing apoptosis [103]. Doxycycline (DOX) induces mitochondrial autophagy and promotes TGEV replication in IPEC-J2 cells [104]. Autophagy is also induced by TGEV infection in PK-15 and swine testicular cells (STs), but autophagy negatively regulates TGEV replication [105].

PEDV infects porcine intestinal epithelial cells (IPEC-J2), African green monkey kidney cells (Vero) and porcine kidney (PK) cells. The proteomic results of Vero cells infected by PEDV show that the expression of microtubule-associated protein 1B (an autophagy marker protein) is up-regulated [106]. However, the mammalian target of the rapamycin (mTOR) pathway is down-regulated, and the expression of autophagy-associated protein 5 (ATG5) is up-regulated [107], indicating that PEDV infection promotes autophagy in Vero cells. The increase in double-membrane vesicles (DMVs) in Vero cells during PEDV infection indicates that Vero triggers autophagy to promote PEDV replication, and autophagy is positively correlated with the PEDV-induced NF-κB pathway [108]. A study has shown that PEDV-induced autophagy enhances replication through PI3K/Akt/mTOR signal pathway in IPEC cells [109]. On the contrary, rapamycin increases autophagy flux in IPEC-J2 cells, inhibits PEDV infection, and reduces PEDV-induced cell death [110]. Another study shows that bone marrow stromal cell antigen 2 (BST2) can recruit E3 ubiquitin ligase to deliver N proteins to autophagosomes through the ubiquitin pathway for selective degradation and inhibit the replication of PEDV in LLC-PK1 and Vero cells [111].

The transcriptome results of SADS-CoV in Vero-E6 cells show that SADS-CoV infection down-regulates the expression of PI3K and AKT genes in the autophagy-related PI3K/Akt/mTOR signal pathway, suggesting that autophagy plays a proviral role in SADS-CoV infection [112]. The relationship between SADS-CoV and autophagy needs to be further studied due to the short time of discovery.

The first proteomic analysis of PDCoV infection in IPEC-J2 cells shows that the autophagy-related pathway PI3K/AKT/mTOR is activated, suggesting that PDCoV infection induces autophagy in IPEC-J2 cells [113]. PDCoV infection also increases the amount of DMVs in PK cells, and the ratio of LC3-II/LC3-I. Additionally, the degradation of SQSTM-1 (p62) is detected, as an indicator of autophagy, indicating that PDCoV may induce autophagy, such as other SeCoVs [114]. The P38MAPK signal pathway is a major cellular signal pathway, which can regulate many cellular responses, including autophagy. Ergosterol peroxide (EP) exhibits the p38MAPK pathway and PDCoV replication in LLC-PK1 cells, suggesting that PDCoV may enhance infection by inducing autophagy [115], and the present study proves this conclusion [116].

Nsp6 of other types of CoV (such as MHV, SARS-CoV, IBV) can stimulate omega intermediates to produce autophagosomes in the endoplasmic reticulum [1], so nsp6 may also be the critical factor of autophagy induced by SeCoV.

The differences in the relationship between SeCoVs and autophagy may be due to the differences between cell lines and virus strains, calling for more comprehensive studies in vivo. Vero is an IFN-deficient cell line, so the competition between the virus and the host may benefit the virus. In order to determine the relationship between SeCoV infection and autophagy at the cellular and molecular levels, IPEC-J2, which is very similar to natural target cells, is a better choice.

## 4. Apoptosis Induced by SeCoV Infection

Apoptosis is a ubiquitous mode of cell death, also known as programed cell death, and an energy-dependent process strictly regulated by genes [117]. Morphological changes of apoptotic cells, including cell membrane atrophy and deformation, chromatin condensation, nuclear fragmentation, plasma membrane blistering, and the formation of apoptotic bodies [118]. Unlike apoptosis, cell necrosis is not controlled by genes, and the release of cytoplasmic contents into the extracellular space can induce inflammation.

There are two pathways of apoptosis: the extrinsic pathway mediated by mitochondria and intrinsic pathway mediated by death receptors (DRs). Each pathway activates specific pro-cysteinyl aspartic acid protease (caspase), eventually inducing apoptosis (Figure 5). The change of mitochondrial outer-membrane permeability (MOMP) plays an essential role in the intrinsic apoptosis pathway, which is coordinated by B-cell lymphoma 2 (Bcl2) family proteins [119]. BAX and BAK are pro-apoptotic proteins that increase MOMP, whereas Bcl2-like proteins are anti-apoptotic proteins that inhibit the increase in MOMP. Under stressful conditions, BAX or BAK translocate into the mitochondria and combine with the BH3 domain of Bid to destroy the integrity of the mitochondrial membrane. The increase in MOMP promotes the release of cytochrome c (Cyt *c*), contributing to the oligomerization of APAF-1 (apoptotic protease-activating factor-1), and then binds to the precursor of caspase-9 to form an apoptotic complex. Subsequently, activated caspase-9 cleavage downstream related caspase family proteins (such as caspase-3, caspase-6, and caspase-7) to induce apoptosis [120,121]. DRs belong to the tumor necrosis factor receptor (TNFR) superfamily in the extrinsic pathway. The binding of death ligand (such as the fatty acid synthase ligand (FasL) and TNF-α Apo3 ligand (Apo3L)) DRs leads to the formation of the death-induced signal complex (DISC) and the activation of caspase-8 to initiate the executive stage of apoptosis [121,122].

The relationship between virus and apoptosis is bidirectional. Viruses can increase virus production or release offspring viruses to infect other cells by delaying apoptosis or promoting apoptosis [123]. TGEV infects PK-15 and ST cells to induce apoptosis (Table 1), in which p53 and the reactive oxygen species (ROS)-mediated apoptosis-inducing factor (AIF) pathway and caspase-dependent pathway are involved [124,125]. TGEV infection induces ROS accumulation in PK-15 cells and decreases the mitochondrial membrane potential. ROS is involved in the activation of p38MAPK and p53, and p53 can, in turn, positively regulate the level of ROS, resulting in apoptosis [126]. In addition, TGEV infection down-regulates Bcl2, up-regulates the expression of BAX, promotes BAX translocation from cytoplasm to mitochondria, activates the mitochondrial-mediated intrinsic apoptosis pathway leading to the release of Cyt *c*, and then activates caspase-9 [127]. TGEV infection can also up-regulate FasL and activate its mediated extrinsic apoptosis pathway, eventually activating caspase-3 and the cleavage of poly (ADP-ribose) polymerase (PARP), leading to apoptosis [127]. These researches suggest that TGEV infection can regulate apoptosis through extrinsic and intrinsic pathways. On the contrary, TGEV may not induce apoptosis of intestinal cells in piglets [128]. The TGEV N protein can locate to the nucleolus and may destroy the cell cycle [129]. The TGEV N protein up-regulates p53 and p21 in S phase and G2max M, thus inhibiting the expression of B1, cdc2, and cdk2 and promoting the translocation of BAX from the cytoplasm to mitochondria, resulting in the release of Cyt *c* and the activation of caspase-3, finally resulting in the apoptosis of PK-15 cells [130]. TGEV infection induces mitochondrial autophagy and up-regulates antioxidant genes, such as DJ-1, to inhibit apoptosis and promotes TGEV replication in IPEC-J2 cells [103].

PEDV infection can damage the integrity of intestinal barrier and lead to the death of piglets, and apoptosis may be the critical factor in the pathogenicity of PEDV. AIF translocation to nucleus during PEDV infection induces the apoptosis of Vero cells, without caspase dependence, and the inhibition of AIF with Cyclophilin D (CypD) can inhibit PEDV infection [123]. Activated p53 and accumulated ROS play an essential role in the apoptosis of Vero cells mediated by caspase-3 and caspase-8 during PEDV infection, and p38 MAPK is not involved in the process of apoptosis [131,132]. In addition, IPEC-J2 cells lacking lymphotoxin beta receptor (LTβR) are more susceptible to apoptosis caused by PEDV infection, suggesting that LTβR may be a natural anti-apoptotic gene [133]. The studies on apoptosis induced by the PEDV protein show that the S1 protein is the most critical functional protein in inducing apoptosis [134]; on the contrary, the ORF3 protein promotes proliferation by inhibiting apoptosis [135].

As recently discovered SeCoVs, PDCoV, and SADS-CoV are related to apoptosis through limited studies, it is reported that SADS-CoV infection can induce the apoptosis of Vero and IPI-2I cells in vitro and the apoptosis of ileal epithelial cells in vivo. SADS-CoV infection up-regulates FasL and induces apoptosis through the cascade reaction of caspase-8 and caspase-3 [136]. In addition, Bid translocates to mitochondria after cleavage by caspase-8, destroying the integrity of mitochondria to promote the release of Cyt *c*, which, in turn, activates caspase-9 and promotes apoptosis [136]. SADS-CoV infection triggers Bax recruitment into mitochondria, resulting in the release of MOMP and Cyt *c*, resulting in cell death. The results suggest that SADS-CoV induces apoptosis through extrinsic and intrinsic pathways, caspase-dependent. Apoptosis inhibition restricts virus replication, indicating that SADS-CoV promotes self-replication by inducing apoptosis; the exact viral proteins involved in apoptosis are still unknown. PDCoV infection induces BAX recruitment into mitochondria, triggers MOMP, the opening of the mitochondrial permeability transition pore (MPTP) and the release of Cyt *c*, leading to the apoptosis of ST cells through the intrinsic pathway [137]. PDCoV infection in vitro can cause apoptosis in LLC-PK cells. In vivo infection can not cause apoptosis in intestinal cells, instead leading to necrosis [138].

The current research results show that SeCoVs utilize complex strategies to regulate apoptosis in different periods to promote replication. Infections in different cells activate different apoptosis pathways, reflecting the complex interactions between the virus and host cells. However, there are still many problems to be solved, such as the fact that the mechanism of apoptosis induced by viral proteins is not clear, the results of virus infection in vitro and in vivo are not consistent, and a better infection model is needed.

## 5. SeCoV and the Innate Immune Mechanism

Innate immunity is a conservative immune strategy important for pathogen identification, restriction, and the subsequent activation of adaptive immunity, considered as the first line of defense against pathogens [139].

### 5.1. Pattern Recognition Receptors

Host cells first sense the presence of viruses and virus products with pathogen-related molecular patterns (PAMPs) through different types of pattern recognition receptors (PRR) after virus infection, triggering different signal transduction pathways and finally activating related immune pathways to inhibit virus replication. The PRRs for virus recognition mainly include endosome Toll-like receptors (TLRs), cytoplasmic RIG-I-like receptors (RLRs), and nuclear oligomeric domain (NOD)-like receptors (NLRs) [140].

TLR is a type I transmembrane protein, which exists in the cell and on the surface of the cell membrane. TLRs mediate the recognition of PAMP and damage-associated molecular patterns (DAMPs) from various sources (including bacteria, fungi, and viruses) through leucine-rich repetitive (LRR) sequence domains [141]. The activation of TLRs mainly occurs in antigen-presenting cells (APCs), including dendritic cells (DCs), monocytes, and B cells.

According to the cell location and PAMP ligand, the functions of TLRs are different. The TLR3, 7, 8, and 9 expressed on the vesicles are involved in recognizing microbial nucleic acids in the vesicles [142]. TLR3 detects double-stranded RNA (dsRNA) viruses (e.g., reovirus) and single-stranded RNA (ssRNA) viruses that produce dsRNA during replication (e.g., West Nile virus) and some small interference RNA (siRNA) [143]. TLR7 and 8 detect ssRNA, while TLR9 mainly detects DNA viruses [144,145]. Other types of TLRs expressed on the cell surface, such as TLR2 and 4, recognize viral proteins, such as hepatitis virus and human immunodeficiency virus (HIV) [146,147]. Different TLR induces specific biological effects. For example, the activation of TLR3 and 4 not only induce IFN-I, but also induce the expression of inflammatory cytokines, while TLR1, TLR2, TLR6, and TLR5 mainly mediate the production of inflammatory cytokines. After activation, TLRs recruit signal transduction molecules containing Toll/IL-1 receptors (TIR) to activate specific signal pathways, such as NF-κB and MAPK signaling pathways activated by the myeloid differentiation primary response protein 88 (MyD88) [148,149].

RLRs are a type of pattern recognition receptor located in the cytoplasm, belonging to the DExD/H-box RNA helicase family (x can be any amino acid). RLRs contain three members: RIG-I (also known as DDX58), melanoma differentiation-associated factor 5 (MDA5, also known as helicard or IFIH1), and laboratory of genetics and physiology 2 (LGP2) [150,151]. Both RIG-I and MDA5 contain N-terminal caspase-recruitment domains (CARD), binding to the CARD domain of downstream junction proteins. In addition, RIG-I and MDA5 contain a DExD/H-box RNA helicase domain that binds to RNA and a repressor domain (RD) located at the C terminal. However, LGP2 lacks an N-terminal CARD domain [152], positively or negatively regulating RIG-I and MDA5, considered as the inhibitor or cofactor of the RLR signaling pathway [153,154,155]. RIG-I recognizes the short dsRNA in the virus genome containing the stalk-like structure of 5′-triphosphate, while, in contrast, MDA5 recognizes long dsRNA fragments [156,157,158]. Conformational changes occur when RIG-I and MDA5 bind to dsRNA, which releases the N-terminal CARD domain and finally activates the MAVS, also known as IPS-1, VISA, or CARDIF. Then, MAVS induces the activation of transcription factors, such as the translocation of NF-κB into the nucleus, to initiate the expression of IFN-I and pro-inflammatory cytokines [159,160,161]. 

NLRs are cytoplasmic PRRs with typical conserved domains: the CARD or Pyrin domain (PYD) at the N terminal, NOD motif in the middle, and LRR motif at the C terminal [162]. LRR motifs detect PAMP and induce molecular conformational rearrangement, eventually triggering oligomerization through NOD to activate signaling pathways, including the NF-κB and MAPK pathways [163]. Subsequently, the associated NLRs oligomize into inflammatory bodies to activate caspase-1, such as the NLR family PYD-containing 1 (NLRP1) and NLRP3. The activated caspase-1 then cleaves the family of pro-inflammatory interleukin-1 (IL-1) cytokines into active forms IL-1β and IL-18 [164].

### 5.2. Interferon and Innate Immunity

IFNs are divided into three types: type I, II, and III. Type I IFN with antiviral activity is an essential aspect of innate immunity. Type I IFN works through the heterodimer formed by IFN-α receptor 1 (IFNAR1) and IFNAR2, then triggers the phosphorylation of JAK1 and tyrosine kinase 2 (TYK2), which leads to the phosphorylation of intracellular tyrosine residues by specific receptors, resulting in phosphorylation of STAT1 and 2. Then, STAT1 and 2 dimerization recruit IRF9 to form IFN-stimulated gene factor 3 (ISGF3) [165]. ISGs exert antiviral effects from diverse aspects. For example, tripartite-motif 5α (TRIM5α) directly binds to and destroys the nucleocapsid protein of the retrovirus, and then releases viral PAMPs into the cytoplasm, promoting an antiviral response after detection by PRRs [166].

### 5.3. Interactions between SeCoV and the Innate Immunity

Interferon levels decreased significantly in patients with severe SARS-CoV and MERS-CoV infection, and coronavirus replication was significantly inhibited after treatment with IFN-α [161], suggesting that type I IFNs played an essential role in the antiviral effect against CoVs infection.

IFN system cannot completely inhibit CoVs infection, notwithstanding the possession of solid antiviral activity. SeCoVs have evolved various strategies to antagonize the interferon pathway, so that the viruses can spread more effectively between host cells [167,168,169]. Based on the studies of PEDV-infected cells, two mechanisms have been summarized to explain the inhibition of type I IFN production mediated by SeCoV infection. Firstly, the virus genomic and subgenomic RNA evade PRRs recognition through 5′ cap methylation [167,170,171]. Secondly, the virus-encoded proteins interfere with the innate immunity-related pathways [172,173]. The structural, non-structural, and accessory proteins of SeCoVs have been proven to change the innate immune response (Figure 6).

### 5.4. Viral Proteins Related to Innate Immunity

#### 5.4.1. Structural Proteins of SeCoVs 

TGEV M and E proteins play a significant role in inducing IFN-α (Table 1). Cells co-expressing M and E proteins induce IFN-α, almost as effectively as cells infected with TGEV [174]. The PEDV S protein interacts with EGFRs, weakening type I IFN activity through the downstream JAK2-STAT3 signal pathway, thus enhancing PEDV replication [64]. The ectopic expression of PEDV E, M, and N proteins in MARC-145 cells has been shown to antagonize the activities of IFN-β and IRF3 [173]. N protein also mediates NF-κB activation through the TLR2, TLR3, and TLR9 pathways [175]. Further studies show that the PEDV N protein inhibits the activation of IRF3 and the production of type I IFN by directly interacting with TBK1 [176]. SADS-CoV N protein inhibits IFN-β production by targeting TBK1 to interfere with TIR domain-containing adaptor inducing interferon-beta (TRAF3) and TBK1 [177]. The SADS-CoV N protein also interacts with RIG-I and mediates the ubiquitination of K27-, K48-, and K63-connections of RIG-I to antagonize IFN-β production [178]. The PDCoV N protein inhibits IFN-β production by interfering with dsRNA binding and the protein activator of protein kinase R (PACT) to RIG-I to inhibit the K63 polyubiquitin of RIG-I [169,179]. It can be concluded that different SeCoVs use different forms of viral structural proteins to affect the innate immune signal pathways and may also interact with various host signal molecules, relating to the different host adaptability of SeCoVs.

#### 5.4.2. Non-Structural and Accessory Proteins of SeCoVs 

Nsps of CoVs have been proved to possess the function of participating in viral RNA synthesis [180,181,182], inhibiting innate immunity to create opportunities for virus invasion and replication. TGEV nsp14 plays an essential role in the modulation of innate immunity, with its role as a PAMP [183]. Recent studies have shown that nsp14 induces IFN-β production in PK-15 cells, and nsp14 interacts with the DDX1 of DExD/H helicase family to induce IFN-β production through the NF-κB pathway [184]. PEDV nsp1 interferes with IRF and blocks type I and type III interferon-mediated production by NF-κB [173,185,186]. PEDV nsp1 also affects the production of proinflammatory cytokines, such as TNF-α, IL-1 β, IL-6, IL-15, and IL-17 [185]. The PLpro domain of PEDV nsp3, papain-like protease 2 (PLP2), has been shown to strongly inhibit the expression of type I IFNs activated by RIG-I and the stimulator of interferon genes (STING), utilizing de-ubiquitination activity to antagonize the signal transduction mediated by RIG-I and STING [187]. 3C-like protease (3CLpro) encoded by the nsp5 gene negatively regulates innate immunity in PEDV infection. PEDV nsp5 proteolysis cleaves the NF-κB essential modulator (NEMO) at Q231, which antagonizes the production of type I IFN and the activation of the downstream RIG-I/MDA5 signal pathway [188]. SADS-CoV nsp1 significantly inhibits the phosphorylation of STAT1-S727, interfering with the effect of type I IFN, and this phenomenon may exist in all *alphacoronavirus* [189]. Similar to PEDV nsp5, PDCoV nsp5 blocks the production of IFN-β by reducing the expression of NEMO [190], and also cleaves the porcine mRNA-decapping enzyme 1a (pDCP1A) to decrease the antiviral activity of pDCP1A [191]. In addition, PDCoV nsp5 hinders the antiviral function of ISGs by the targeted cleavage of STAT2 [192]. Accessory proteins play a role in the host–pathogen interaction and mediate viral pathogenesis. Accessory proteins encoded by genes, such as ORF3, ORF6, and ORF9 in SARS-CoV and ORF4a/b in MERS-CoV, have immunomodulatory activity [193,194,195]. TGEV ORF7 inhibits the phosphorylation of eukaryotic translation initiation factor 2 (eIF2α) and the activation of RNase L by binding to the protein phosphatase 1 catalytic subunit (PP1c), antagonizing the host antiviral response [196]. As the only accessory protein, PEDV ORF3 has been shown to inhibit the induction of type I IFN in vitro [173]. PDCoV NS6 interacts with RIG-I and MDA5, thus interfering with the combination of RIG-I/MDA5 with dsRNA to inhibit the production of IFN-β [197]. The host also has corresponding countermeasures: after TGEV infection, microRNA-4331 (miR-4331) directly targets the cell division cycle-associated protein 7 (CDCA7) to inhibit ORF7 transcription [198].

## 6. Concluding Remarks

SeCoV has been threatening the farming industry as an important type of pathogen in the past decades, but the mechanism of its interactions with the organism needs further study. SeCoV tends to utilize various components of host cells to promote replication and host pathogenesis, and hosts set up various mechanisms to control and resist viral infection. Due to the COVID-19pandemic, the phenomenon of cross-species transmission of SeCoV is also a noteworthy research direction, such as the discovery of PDCoV in Haitian children. The association of SeCoV with bats and birds may be the key to studying the origin and transmission of SeCoV. Besides, there are still many unknown questions about SeCoV, such as “what are the receptors for PEDV, PDCoV, and SADS-CoV?”. In-depth research on all aspects of SeCoV is an exciting area and will help to develop effective antiviral drugs and vaccines.

## Figures and Tables

**Figure 1 ijms-23-03953-f001:**
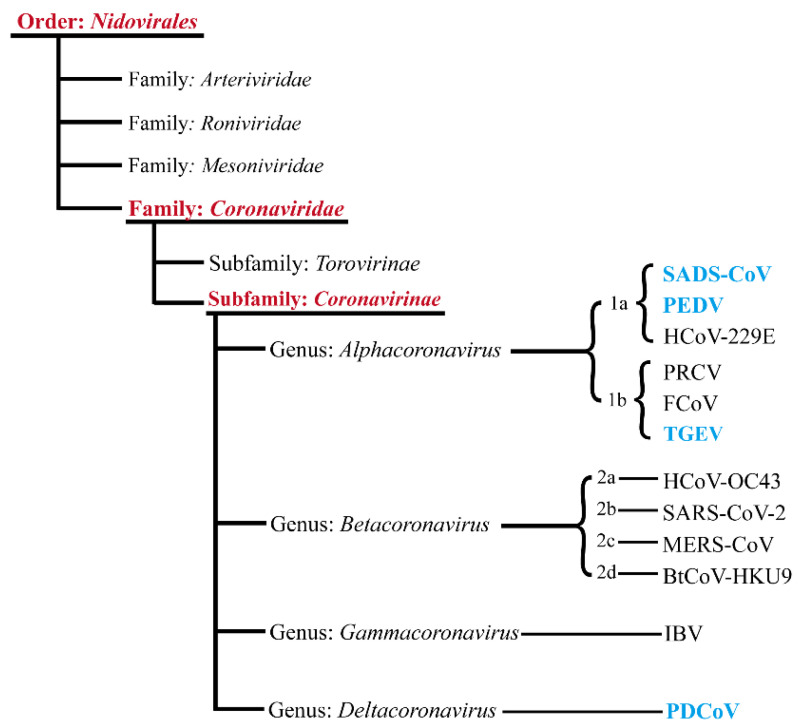
Taxonomy of SeCoVs. Schematic diagram showing the classification of coronaviruses. The four known SeCoVs are marked in blue in the figure. Abbreviations: BtCoV, bat coronavirus; IBV, infectious bronchitis virus; and PRCV, porcine respiratory coronavirus.

**Figure 2 ijms-23-03953-f002:**
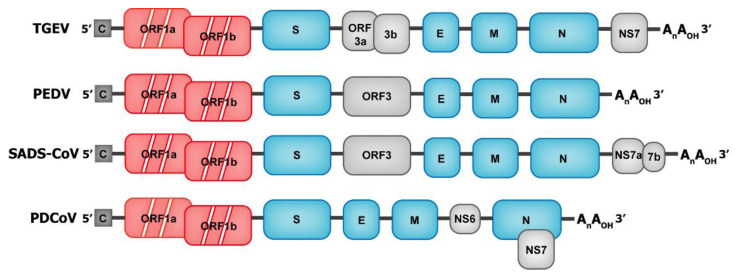
Genome structure of swine enteric coronaviruses (SeCoVs). The genomic structures of the four known SeCoVs are shown in the schematic diagram above (not to scale). The 5′-cap structure and 3′ polyadenylation are 5′-C and AnAOH-3′, respectively. Following the open reading frame ORF1ab (marked in red) are genes encoding structural and accessory proteins. Structural proteins (S, E, M, and N) are marked in blue and accessory proteins are marked in gray.

**Figure 3 ijms-23-03953-f003:**
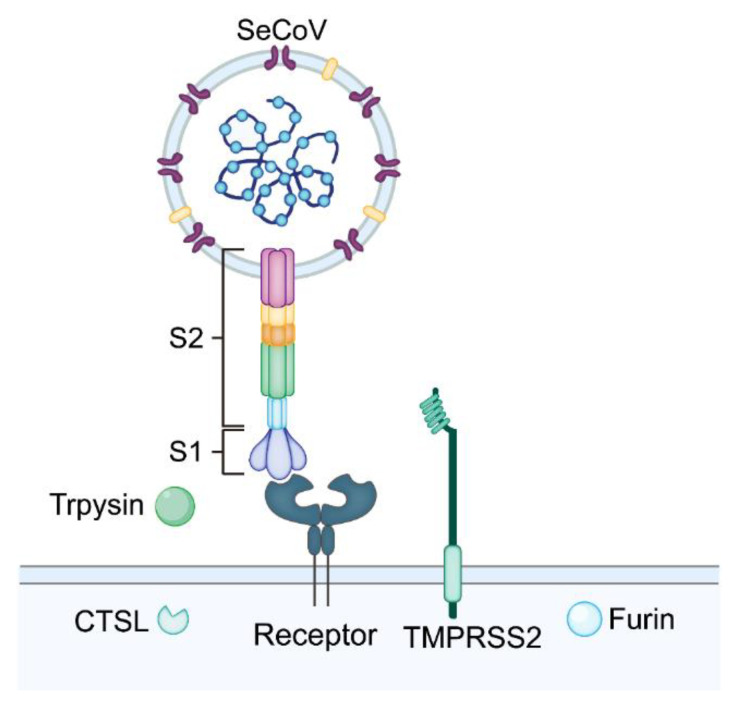
Attachment and entry of SeCoV. Schematic diagram showing the attachment and entry processes assisted by the host proteases of SeCoV. The S1 region of SeCoV’s S protein binds to specific receptors, and the S2 region helps the virus integrate with the cell surface. The host proteases cleave the S protein to facilitate membrane fusion and allow the virus to enter the cell.

**Figure 4 ijms-23-03953-f004:**
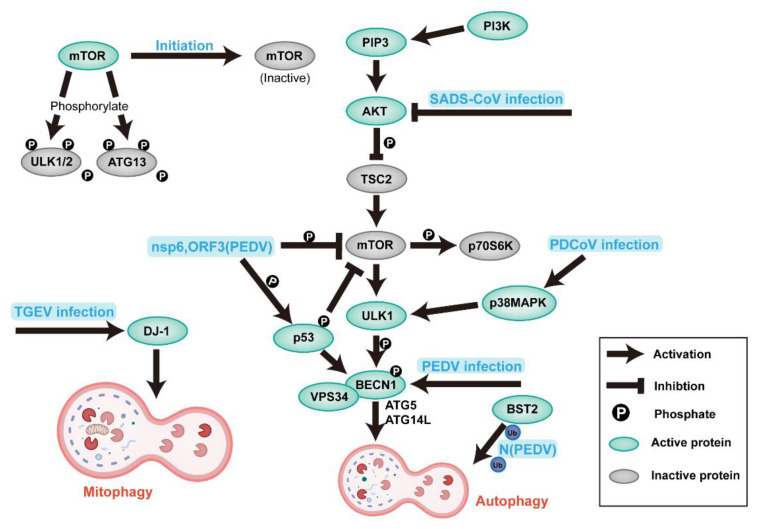
Induction and modulation of host autophagy by SeCoV infection. Schematic diagram showing the organism’s signaling pathways in autophagy and the regulatory mechanisms utilized during SeCoV infection. The viral or viral components involved in the modulation of autophagy are shown in bold blue in the figure. Abbreviations: ULK1/2, Unc-51-like autophagy-activating kinase1/2; ATG, autophagy-related gene; PIP3, phosphatidylinositol 3,4,5-trisphosphate; TSC2, tuberous sclerosis complex 2; p70S6K, ribosomal S6 protein kinase; p53, tumor protein 53; VPS34, vesicular protein sorting 34; DJ-1, deglycase 1; BECN1 (beclin1), coiled-coil myosin-like Bcl2-interacting protein; and BST2, bone marrow stromal cell antigen 2.

**Figure 5 ijms-23-03953-f005:**
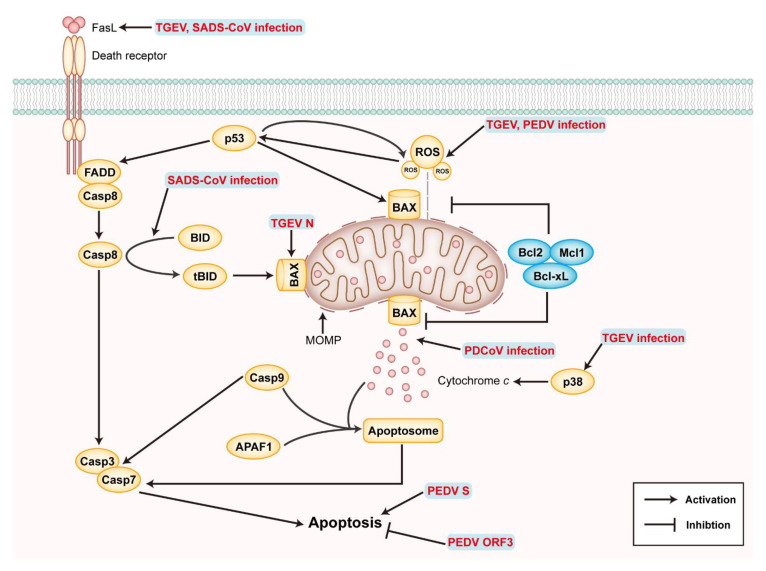
Apoptosis induced by SeCoV infection and modulatory mechanisms. Schematic diagram showing the signaling pathways of intrinsic and extrinsic apoptosis and the modulatory mechanisms during SeCoV infection. The yellow ovals are proapoptotic proteins, whereas the blue ovals are antiapoptotic proteins. The viral or viral components involved in the modulation of apoptosis are shown in bold red in the figure. Abbreviations: FADD, Fas-associated via death domain; BID, BH3-interacting domain death agonist; BAX, Bcl2-associated X; Mcl1, myeloid cell leukemia 1; Mcl1, myeloid cell leukemia 1; and APAF1, apoptotic peptidase-activating factor 1.

**Figure 6 ijms-23-03953-f006:**
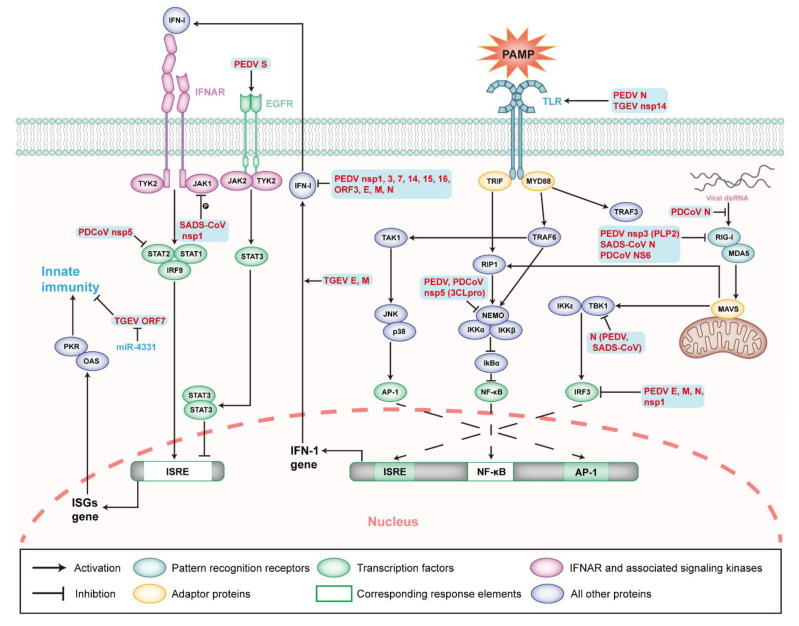
Innate immunity during SeCoV infection and modulatory mechanisms. Schematic diagram showing the type I interferon-induced innate immunity signaling pathways and known modulatory mechanisms during SeCoV infection. The viral or viral components involved in the modulation of apoptosis are shown in bold red in the figure. Abbreviations: PKR, protein kinase RNA-activated; OAS, 2′5′-oligoadenylate synthetase; ISRE, interferon-stimulated response element; AP-1, activator protein 1; TRIF, TIR domain-containing adaptor inducing interferon-beta; RIP1, receptor-interacting serine/threonine kinase 1; IKK, IkappaB-kinase; and IκBα, NF-κB inhibitor alpha.

**Table 1 ijms-23-03953-t001:** Information on SeCoVs.

	TGEV, PRCV	PEDV	SADS-CoV	PDCoV
Genus	*Alphacoronavirus*			*Deltacoronavirus*
Genome	5′UTR-ORF1a/1b-S-ORF3a/3b-E-M-N-NS7-3′UTR	5′UTR-ORF1a/1b-S-ORF3-E-M-N-3′UTR	5′UTR-ORF1a/1b-S-ORF3-E-M-N-NS7a/7b-3′UTR	5′UTR-ORF1a/1b-S-E-M-NS6-N-NS7-3′UTR
First discovered	1933 (TGEV) 1984 (PRCV)	1978	2017	2009
First reported	U.S.A., 1946 (TGEV) Belgium, 1984 (PRCV)	U.K., 1971	Guangdong, China,2017	U.S.A., 2014
Disease distribution	America, Europe, Asia, Africa	America, Europe, Asia	China	U.S.A., China, Thailand
Clinical symptoms	Diarrhoea, dehydration, weight loss, death Dyspnea, tachypnea, sneezing, coughing, fever (PRCV)			
Mortality	Approaching 100% in piglets less than 2 weeks old	About 50–90% in suckling piglets	Up to 90% for piglets ≤ 5 days of age and up to 5% for pigs over 8 days old	Up to 40% in neonatal piglets
Morbidity	Less than 3%	80–100%	About 10%	20–30%
Receptor and cofactors	pAPN; TfR1, EGFR	Receptor is unknown; SA, EGFR, Neu5Ac	Unknown	Receptor is unknown; SA
Autophagy	Induces mitophagy	Activates p53 and inhibits mTOR pathways	Inhibits PI3K/AKT/mTOR pathway	Activates p38MAPK pathway
Apoptosis	Extrinsic and intrinsic pathways			Intrinsic pathway
Innate immunity	E and M proteins induce IFN-α; ORF7 inhibits innate immunity; Nsp14 activates NF-κB pathway	Activates JAK2-STAT3 and NF-κB pathways; N and nsp1 inhibit TBK1-IRF3 pathway; Nsp3 inhibits RIG-I-MAVS pathway	Nsp1 inhibits JAK1-STAT1 pathway; N inhibits TBK1-IRF3 pathway	N inhibits RIG-I-MAVS pathway; Nsp5 activates NF-κB pathway

Abbreviations: pAPN, porcine aminopeptidase N; TfR1, transferrin receptor 1; EGFR, epidermal growth factor receptor; SA, sialic acid; Neu5Ac, N-acetylneuraminic acid; mTOR, the mammalian target of rapamycin; PI3K, phosphatidylinositol 3-kinase; AKT, RAC-alpha serine/threonine protein kinase; p38MAPK, p38 mitogen-activated protein kinase; JAK, Janus kinase; STAT, signal transducers and transcript; RIG-I, retinoic acid-inducible gene I; MAVS, mitochondrial antiviral signaling; TBK1, TANK-binding kinase 1; and IRF3, IFN regulatory factor 3.

## Data Availability

Not applicable.

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
