# Peer review of "Swine Enteric Coronavirus: Diverse Pathogen–Host Interactions"

_ijms, 2022, doi:10.3390/ijms23073953_

Round 1

Reviewer 1 Report

Porcine coronaviruses have strong impact on the economics of many countries. Currently, there are not many effective commercial vaccines against porcine coronaviruses, which further increase the negative effect on pig husbandry. In addition, the COVID-19 epidemic showed the possibility of interspecies transmission of coronaviruses, which makes their study an urgent task.

The article is a fairly detailed review of coronavirus diseases in pigs, including the mechanisms of interaction between the virus and host cells. The work leaves a good impression, the authors analyzed a large amount of information.

At the same time, I have several comments:

Line 75 – please, add Genbank accession numbers for HCoV-229E, HCoV-NL63 strains, if it possible.

Line 217-220 – I think the sentence structure here is not very good. It's too long and complicated and doesn't sound very clear.

Line 293-294 –From this sentence, the reader can conclude that autophagy always plays an antiviral role in the infection with a DNA virus. But this is not entirely true. According to the data presented in the work of Yin, H. C., et.al, (2019) (DOI: 10.3390/v11090776), many DNA viruses are able to use autophagy for their own purposes. Perhaps this section should be expanded a bit to provide more examples of the various roles of autophagy in viral infection.

Line 623 - Default text from Acknowledgements section should be removed.

In addition, there is information in the literature about the presence of the antibody-dependent enhancement phenomenon in some coronaviruses. In particular, ADE has been well studied in feline enteric coronavirus. There are many publications on the study of ADE in SARS-CoV2. However, I have not seen any information about the ADE for porcine coronaviruses. As a suggestion to the authors, in the introduction, ADE in porcine coronaviruses could be mentioned if there is any information about it. Or indicate that this effect has not been studied for porcine coronaviruses.

Author Response

Dear Reviewer:

Thank you for your valuable comments and suggestions. Based on your comments and suggestions, we have made the following changes:

Reply:

Line 82 - Genbank accession numbers for HCoV-229E (GenBank: KU291448) and HCoV-NL63 (GenBank: LC687394) have been added. 

Lines 224-226 - Sentence has been rewritten. - "Exogenous expression of pAPN can successfully infect PDCoV in non-infectious cell lines, reflecting the origin of PDCoV and its mechanism of cross-species transmission."

Lines 298-301 - Added the fact that porcine circovirus type 2 (PCV2) and certain other DNA viruses use autophagy to facilitate replication. - "Autophagy plays an antiviral role during herpesvirus HSV-1 infection, while adenovirus, porcine circovirus type 2 (PCV2) and certain other DNA viruses can use autophagy to promote replication."

Lines 608-609 - The default text for the Acknowledgments section has been removed.

Lines 33-37 - Added ADE phenomenon for FCoV and SARS-CoV-2, noting that there are no ADE studies on SeCoV. - "It is worth noting that certain coronaviruses are now known to trigger antibody-dependent enhancement (ADE) to enhance infection, such as SARS-CoV-2 and feline coronavirus (FCoV). Studies on the presence of enteroviruses in the swine gut. Coronavirus (SeCoV) can trigger ADE because relatively little research has been done on SeCoV.” Thanks again for your suggestion.

Reviewer 2 Report

This is an ambitious attempt that authors try to summarize the up-to-date facts regarding 4 members of the swine enteric coronaviruses (SeCoV), namely TGEV including PRDC, PEDV, PDCoV, SADS-CoV.  In order for this manuscript to become a more valuable reference for veterinary scientists as well as for basic scientists, I would suggest the following revisions.

General comments:

  • There are too many words. Authors have cited much details from literature but readers may lose the key points along the reading. I suggest reducing the volume by omitting the names of authors (references) and by not reciting the experimental conditions. Just write down the current understanding on each topic and cite the reference number. Use the present tense as much as possible.
  • There are too many references. I suggest avoid citing those literature that were published before year 2000 or those facts that are already stated in textbooks, unless necessary, for example to mention the year of first discovery etc.
  • Since these 4 member viruses have similarities as well as differences, I would suggest creating a big table to summarize as follows:

TGEV, PRDC

PEDV

PDCoV

SADS-CoV

genus

alphacoronavrus

deltacoronavirus

genome

(differences)

cite major differences shown in Figure 2.

first discovered (virus)

2008, for example

first discovered (disease and place)

2010, Guanzhou, China for example

disease distribution (geographically)

clinical symptoms

including clinical signs, said diarrhea or respiratory as well, in particular the different ages (suckling, 5-7 days old, weaners, fattener, adults) of pigs most affected or less affected; morbidity, mortality.  These would be particular interested for veterinarians for doing clinical “differential diagnoses” before laboratory diagnosis are conducted. To distinguish among these 4 viral diseases, based on the clinical parameters are most important.

receptor and co-factors

summarize from section 2.2

authophagy

yes or no,

mention step(s) in the signaling pathway that each virus chips in (cite from Figure 4 and section 3)

apoptosis

yes or no,

mention step(s) in the signaling pathway that each virus chips in (cite from Figure 5 and section 4)

innate immunity

mention Step(s) in the signaling pathway that each virus chips in (cite from Figure 6 and section 5)

  • Enlarge Figure 4, and boxed the viruses (the words) wherever they chip into the pathway. Rearrange related text statements in each section according to this figure.
  • Enlarge Figure 5, and boxed the viruses (the words) wherever they chip into the pathway. Rearrange related text statements in each section according to this figure.
  • Enlarge Figure 6, and boxed the viruses (the words) wherever they chip into the pathway. Rearrange related text statements in each section according to this figure.

  • Concluding remarks are too long, I suggest at least, delete lines 587-589; most of line 596. Lines 597-608 are too general, I suggest rewrite and add real facts.

Minor comments:

Line 49: “chorionic” is in the placenta. Use other words such as “intestinal epithelium” or “villi enterocyte”.

Line 228: clarify what are the two ways for CoVs to invade cells: what is the way other than “endocytosis”?  Is this the same as “receptor-mediated endocytosis”? What is the second way?

Author Response

Dear reviewer:
Thank you for your valuable comments and suggestions. Based on your comments and suggestions, we make the following changes:
response:

  1. The names of the authors (references) have been omitted and the experimental conditions cited have been reduced.
  2. The ideas mentioned in the manuscript are stated in the present tense.

  3. We Removed unnecessary pre-2000 references.

  4.  

    TGEV, PRCV

    PEDV

    SADS-CoV

    PDCoV

    Genus

    Alphacoronavirus

    Deltacoronavirus

    Genome

    5’UTR-ORF1a/1b-S-ORF3a/3b-E-M-N-NS7-3’UTR

    5’UTR-ORF1a/1b-S-ORF3-E-M-N-3’UTR

    5’UTR-ORF1a/1b-S-ORF3-E-M-N-NS7a/7b-3’UTR

    5’UTR-ORF1a/1b-S-E-M-NS6-N-NS7-3’UTR

    First discovered

    1933 (TGEV)

    1984 (PRCV)

    1978

    2017

    2009

    First reported

    USA, 1946

    (TGEV)

    Belgium, 1984 (PRCV)

    UK, 1971

    Guangdong, China,2017

    USA, 2014

    Disease distribution

    America, Europe, Asia, Africa

    America, Europe, Asia,

    China

    USA, China, Thailand

    Clinical symptoms

    Diarrhoea, dehydration, weight loss, death

    Dyspnea, tachypnea, sneezing, coughing, fever (PRCV)

    Mortality

    Approaching 100% in piglets less than 2 weeks old

    About 50%-90% in suckling piglets

    Up to 90% for piglets ≤ 5 days of age and up to 5% for pigs over 8 days of age

    Up to 40% in neonatal piglets

    Morbidity

    Less than 3%

    80%-100%

    About 10%

    20%-30%

    Receptor and cofactors

    pAPN; TfR1, EGFR

    Receptor is unknown; SA, EGFR, Neu5Ac

    Unknown

    Receptor is unknown; SA

    Autophagy

    Induces mitophagy

    Activates p53 and inhibits mTOR pathways

    Inhibits PI3K/AKT/mTOR pathway

    Activates p38MAPK pathway

    Apoptosis

    Extrinsic and intrinsic pathways

    Intrinsic pathway

    Innate immunity

    E and M pro-teins induce IFN-α; ORF7 inhibits innate immunity; Nsp14 activates NF-κB pathway

    Activates JAK2-STAT3 and NF-κB pathways; N and nsp1 inhibit TBK1-IRF3 pathway; Nsp3 inhibits RIG-I-MAVS pathway

    Nsp1 inhibits JAK1-STAT1 pathway; N inhibits TBK1-IRF3 pathway

    N inhibits RIG-I-MAVS pathway; Nsp5 activates NF-κB pathway

    Table 1 was created as required; in Figures 3, 4, and 5, the viruses were boxed; Figures 3, 4 and 5 were enlarged; and the order of statements in the relevant paragraphs was rearranged.
  5. Rewrote the Concluding remarks, and related facts were also added.-"SeCoV has been threatening the farming industry as an important type of pathogen in the past decades, but the mechanism of its interactions with the organism needs further study. SeCoV tends to utilize various components of host cells to promote replication and host pathogenesis, and hosts set up various mechanisms to control and resist viral infection. Due to the pandemic of COVID-19, the phenomenon of cross-species transmission of SeCoV is also a noteworthy research direction, such as the discovery of PDCoV in Haitian children. The association of SeCoV with bats and birds may be the key to study the origin and  transmission of SeCoV. Besides, there are still many unknown questions about SeCoV, such as what are the receptors for PEDV, PDCoV and SADS-CoV? In-depth research on all aspects of SeCoV is an exciting area and it will help develop effective antiviral drugs and vaccines."
  6. Line 56: “chorionic” was replaced with "intestinal epithelium" .
  7. Line 234: The two ways for CoVs to invade cells were clarified: 

    "one is endocytosis. Viral S protein interacts with receptor and then virus particles are wraped in endosomes to enters host cells. The other is membrane fusion, through which exogenous proteases or host proteases activate the S protein upon binding of the virus to the receptor, and that virus then enters the cytoplasm through membrane fusion.”

    Thank you again for your suggestions.

Round 2

Reviewer 2 Report

Authors have responded to most of my comments.  I see more coherence between words and figures.  The inclusion of Table 1 gives a bigger picture for the reader and reflects the rumination of the writers.

Still there are a few minor errors that need to be corrected.  For example, in Figure 4, authors found out that the effect of SADS-CoV on AKT is inhibitory (R1 version) rather than stimulatory (original version).  This is possible only if authors try to correlate the figure with the texts.   I therefore urge the authors to check throughout to see if there are similar mistakes. 

line 251: "wrapped" or "wraped"?

line 278: "limit" or "limite"?

Once again, check throughout to correct minor errors.

Author Response

Dear reviewer:
Thank you for your valuable comments and suggestions. Based on your comments and suggestions, we make the following changes:
response:

Line 234 - The word "wraped" was changed to "wrapped";

Line 259 - The word "limite" was changged to "limit"; 

Line 295 - The word "given" was changed to "gives".

Figure 4: The effect of SADS-CoV infection on AKT is indeed inhibitory rather than stimulatory, this error was discovered during the first revision of the manuscript and we have fixed it.

Figure 6: The changes are as follows :

It is "PEDV S" and not "PEDV infection" that interacts with EGFR;

The interactions of "SADS-CoV N" and "PDCoV NS6" with RIG-I were omitted and have been added;

The inhibitory effect of "PEDV E M N" on IRF3 was omitted and has been added.

No errors were found in the remaining Figures and Table 1.

Similar errors like punctuation and word spacing have been fixed.

Thank you again for your responsible and detailed suggestions.